# Mapping translation 'hot-spots' in live cells by tracking single molecules of mRNA and ribosomes

Zachary B Katz[1,2†], Brian P English[1,3†], Timothée Lionnet[3], Young J Yoon[1], Nilah Monnier[4‡], Ben Ovryn[1], Mark Bathe[4], Robert H Singer[1,3*]

[1]Department of Anatomy and Structural Biology, Albert Einstein College of Medicine, New York, United States; [2]Salk Institute for Biological Studies, La Jolla, United States; [3]Janelia Research Campus, Howard Hughes Medical Institute, Ashburn, United States; [4]Department of Biological Engineering, Massachusetts Institute of Technology, Cambridge, United States

*For correspondence: robert.
singer@einstein.yu.edu

[†]These authors contributed
equally to this work

Present address: [‡] Department
of Genetics, Stanford University
School of Medicine, Stanford,
United States

Competing interest: See
page 15

Reviewing editor: Nahum
Sonenberg, McGill University,
Canada

**Abstract** Messenger RNA localization is important for cell motility by local protein translation. However, while single mRNAs can be imaged and their movements tracked in single cells, it has not yet been possible to determine whether these mRNAs are actively translating. Therefore, we imaged single $\beta$-actin mRNAs tagged with MS2 stem loops colocalizing with labeled ribosomes to determine when polysomes formed. A dataset of tracking information consisting of thousands of trajectories per cell demonstrated that mRNAs co-moving with ribosomes have significantly different diffusion properties from non-translating mRNAs that were exposed to translation inhibitors. These data indicate that ribosome load changes mRNA movement and therefore highly translating mRNAs move slower. Importantly, $\beta$-actin mRNA near focal adhesions exhibited sub-diffusive corralled movement characteristic of increased translation. This method can identify where ribosomes become engaged for local protein production and how spatial regulation of mRNA-protein interactions mediates cell directionality.

## Introduction

mRNA compartmentalization is a conserved mechanism cells use to spatially control protein translation through zipcode sequences often in the transcript's 3'UTR (*Kislauskis and Singer, 1992*; *Holt and Bullock, 2009*; *Pratt and Mowry, 2012*). Previous work has shown that MS2 binding sites (MBS) inserted adjacent to the zipcode sequence in the 3'UTR of β-actin mRNA can be used to label and track the movement of single mRNA molecules (*Fusco et al., 2003*; *Yamagishi et al., 2009*; *Lionnet et al., 2011*; *Katz et al., 2012*). Previous studies identified that mRNA: (1) exhibited multiple movement profiles (directed, diffusive, and corralled), (2) zipcode sequences facilitated directed transport of β-actin mRNA along microtubules, and (3) zipcode binding protein 1 (ZBP1) delivered β-actin mRNA to leading edge adhesions where it could dwell for more than a minute. ZBP1 was previously shown to deliver its mRNA target in a translationally silenced state by blocking the 60S ribosomal subunit from initiating translation. Once the RNA-protein complex (RNP) localized to the cell periphery, Src kinase phosphorylated the 396 tyrosine to relax translation repression (*Hüttelmaier et al., 2005*). How translating mRNA can be retained within this compartment is still unclear, although evidence has suggested that the elongation factor 1α (EF1α) may act to anchor mRNA to formins and thereby the leading edge actin cytoskeleton (*Liu et al., 2002*). Accordingly, the actin cytoskeleton was previously shown to be necessary for mRNA localization and maintenance at the leading edge of migrating fibroblasts (*Sundell and Singer, 1991*). A recent report also argued that the RNP size is the main

**eLife digest** The instructions that a cell needs to build proteins are encoded within its genes. To make proteins, these instructions are first copied into molecules of messenger RNA (or mRNA for short). Molecular machines called ribosomes then translate these mRNA molecules to produce proteins. Controlling when and where proteins are produced within a cell is important for tissue development, memory formation in the brain, and the outcome of certain cancers.

Methods that can pinpoint the time and place that translation occurs will lead to an improved understanding of how biological processes take place. While current techniques can be used to track the movement of single mRNA molecules in individual cells, it has not yet been possible to determine whether ribosomes are actively translating these mRNAs.

Katz, English et al. tracked the movements of mRNA molecules (that encode a protein called β-actin) and ribosomes at the same time, by using fluorescent tags with different colors. β-actin is an important component of the cell's internal scaffolding and its localized translation enables cells to move in a given direction. Katz, English et al. confirmed that β-actin mRNA translation most often occurs near the leading edge of a moving cell, especially near sites where the cell attaches to its surroundings to help pull itself forward.

The experiments also revealed that mRNAs that are being translated behaved differently to mRNAs that are not being translated. In particular, translating mRNAs and their associated ribosomes move much slower than non-translating mRNAs. Importantly, this method can be applied to any mRNA of interest, and once the movements of the mRNA have been defined it is possible to estimate whether or not it is being translated. In the future, a similar approach could provide a definitive answer concerning how mRNA molecules interact with proteins within different parts of the cell and will lead to future investigations on localized production of proteins in living cells.

determinant of localization and therefore the architecture of the cytoplasmic microenvironment is a major factor in RNA retention at the leading edge (*Yamagishi et al., 2009*). Our results suggest that delivery to a local site for increased translation may inherently increase β-actin mRNA dwell times and determine cell polarity and consequent directed motility (*Park et al., 2012*).

MBS bound by the GFP-fused coat proteins (MCP) make single molecule imaging of mRNA technically feasible (*Shav-Tal et al., 2004*), but it has not been possible to image translation at the single molecule level. Recently, an approach was developed using binding sites in the coding region that indicated when the first ribosome passed through the mRNA, but this biosensor detected only the pioneer round of translation (*Halstead et al., 2015*). Our goal in this work was to derive an approach that could be used to deduce the likelihood of the translational state at any given time of any fluorescently tagged mRNA. The work presented here describes a method for mapping the quantitative mobility features of a fluorescently tagged mRNA while it is associated with ribosomes to determine whether it is being translated, using multi-spectral live cell imaging. This approach mines data sets consisting of thousands of single mRNA tracks to extract the correlations between mRNA-polysome movements and the cytoplasmic nanoenvironment (for instance, mRNA near focal adhesions were found to move in a more corralled fashion). A novel method for tracking ribosomes simultaneously with labeled mRNA resulted in a high-resolution spatial map of mRNAs actively interacting and moving with ribosomes. mRNAs that travel with ribosomes were likewise found to move slower and be more spatially confined. When translation inhibitors were used to dissociate ribosomes from mRNA, the average diffusion coefficient of β-actin mRNA increased. Therefore it is possible to map regions of active translation based on movement profiles within a given cellular compartment.

## Results

### Single mRNA tracking and mapping reveals that mRNA diffusion depends on cytoplasmic compartment

To investigate mRNA movement based on cytoplasmic location we first established a method to track and map all endogenous β-actin mRNA in live fibroblasts. Constitutive expression of tdMCP-

GFP labeled all endogenous β-actin mRNA derived from a mouse in which 24 MS2 binding sites (MBS) were inserted into both alleles of the *Actb* gene (*Lionnet et al., 2011*). Cells were transfected with paxillin-mCherry to label focal adhesions and spatially distinguish the cell's leading edge (*Figure 1A* and *Figure 1—figure supplement 1A*). We tracked both fast and slow moving mRNAs with TIRF excitation at a frame time of 35 ms (*Video 1*). We used mRNA trajectories with lifetimes beyond 105 ms (three consecutive frames) to build a β-actin mRNA diffusion map, in which all trajectories were given equal weight, regardless of their duration (*Figure 1F* and *1G*, see materials and methods for more details). Spatially averaged mRNA diffusion maps distinguish cell compartments that channel or restrict mRNA movement (*Figure 1F* and *Video 2*). Previous work suggested that a subset of dwelling mRNAs around focal adhesions represented sites of increased translation (*Katz et al., 2012*). In order to verify that we could detect this effect in our assay, we identified the positions of each focal adhesion (*Figure 1D*) and partitioned tracks that localized to adhesion complexes. Diffusion maps revealed that mRNA movement around adhesions was indeed slower (*Figure 1B,C*, and *Figure 1—figure supplement 1B*). A single-diffusive component fit of the cumulative distribution function (CDF) curves determined that β-actin mRNA at adhesions moved on average 37% slower (*Figure 1—figure supplement 2A* vs. *2B*). However, the CDF curves are much better fit by a linear combination of two states, a faster one with an apparent diffusion coefficient of 0.4 $\mu m^2$/s, and a slower one with a coefficient of 0.1 $\mu m^2$/s (*Figure 1B*, *Figure 1—figure supplement 2C,D*). This two-component fit constitutes a straightforward method capable of de-convolving the entire dataset and estimating the percentage of the two mRNA species, fast and slow. Near focal adhesions, the population of fast mRNA species falls from 60% to 50% (*Figure 1B*), which indicates that around focal adhesions mRNAs diffuse slower and dwell longer on average (*Figure 1—figure supplement 1C*). *Figure 1C* depicts the mean square displacement (MSD) curves of adhesion-localized mRNAs, which plateaus at a much lower level. This plateau demonstrates a significant shift towards a corralled movement profile (*Figure 1C*). This local confinement may indicate that β-actin mRNAs are trapped in areas of localized translation around focal adhesions.

## Translating transcripts move slower on average

We next hypothesized that mRNAs would exhibit restricted movement as ribosomes or other RNA binding proteins accumulate on the transcript, thereby increasing its mass and stokes radius. In order to test this, we dissociated ribosomes from β-actin mRNA with puromycin and hippuristanol, both drugs that disrupt mRNA-ribosome interactions (*Joklik and Becker, 1965*; *Bordeleau et al., 2006*; *Darnell et al., 2011*; *Wu et al., 2015*). After treatment, β-actin mRNA trajectories showed a significant shift of movement towards the faster diffusing population (*Videos 3* and *4*). The average CDF from five separate acquisitions after puromycin treatment fit well with the linear combination of two CDFs, but now with ~90% of mRNAs being in the fast population state (*Figure 2A*). Likewise, the MSD curve as well as the histogram of average apparent diffusion coefficients also displayed a significant shift to faster diffusion after puromycin treatment when compared to steady state β-actin mRNA movement (*Figure 2B* and *2C*). This global shift in mRNA diffusion behavior at the leading edge of cells is also apparent when comparing the local apparent diffusion maps between mRNA trajectories without puromycin treatment, as well as after addition of puromycin (*Figure 2D* vs. *2E*). The translation initiation inhibitor hippuristanol showed the same increase in mRNA diffusion after 1 μM treatment for 20 min (*Figure 2—figure supplement 1*, *Video 4*).

To image single ribosomes, the 60S large subunit protein L10A was labeled with PATagRFP for single particle tracking using photo-activated localization microscopy (sptPALM). Stable expression of fluorescently labeled L10A has been shown to label ~40% of ribosomes (*Wu et al., 2015*). Ribosomes could be tracked for an average lifetime of ~200 ms (6.2 frames) after applying 405 nm activation energy. Since a small amount of activation was used in order to track single ribosome molecules, only a small fraction of total ribosomes were tracked. Ribosomes exhibited heterogeneity in diffusive motion, much as with β-actin mRNA (*Video 5*). Ribosome movement in cells at steady-state is best fit with the same two-component fit as mRNA: with a linear combination of a fast apparent diffusion coefficient of 0.4 $\mu m^2$/s, and a slower one with a coefficient of 0.1 $\mu m^2$/s. A two-component fit of experimentally obtained CDFs of ribosome trajectories reveals that ribosomes collectively show a shift towards a slower diffusion component as compared to β-actin mRNA (56% vs. 42%, respectively) and suggests most ribosomes actively engage in translation throughout the cytoplasm (compare *Figure 2A* vs. *2F*). With the addition of puromycin, a significant increase in ribosome

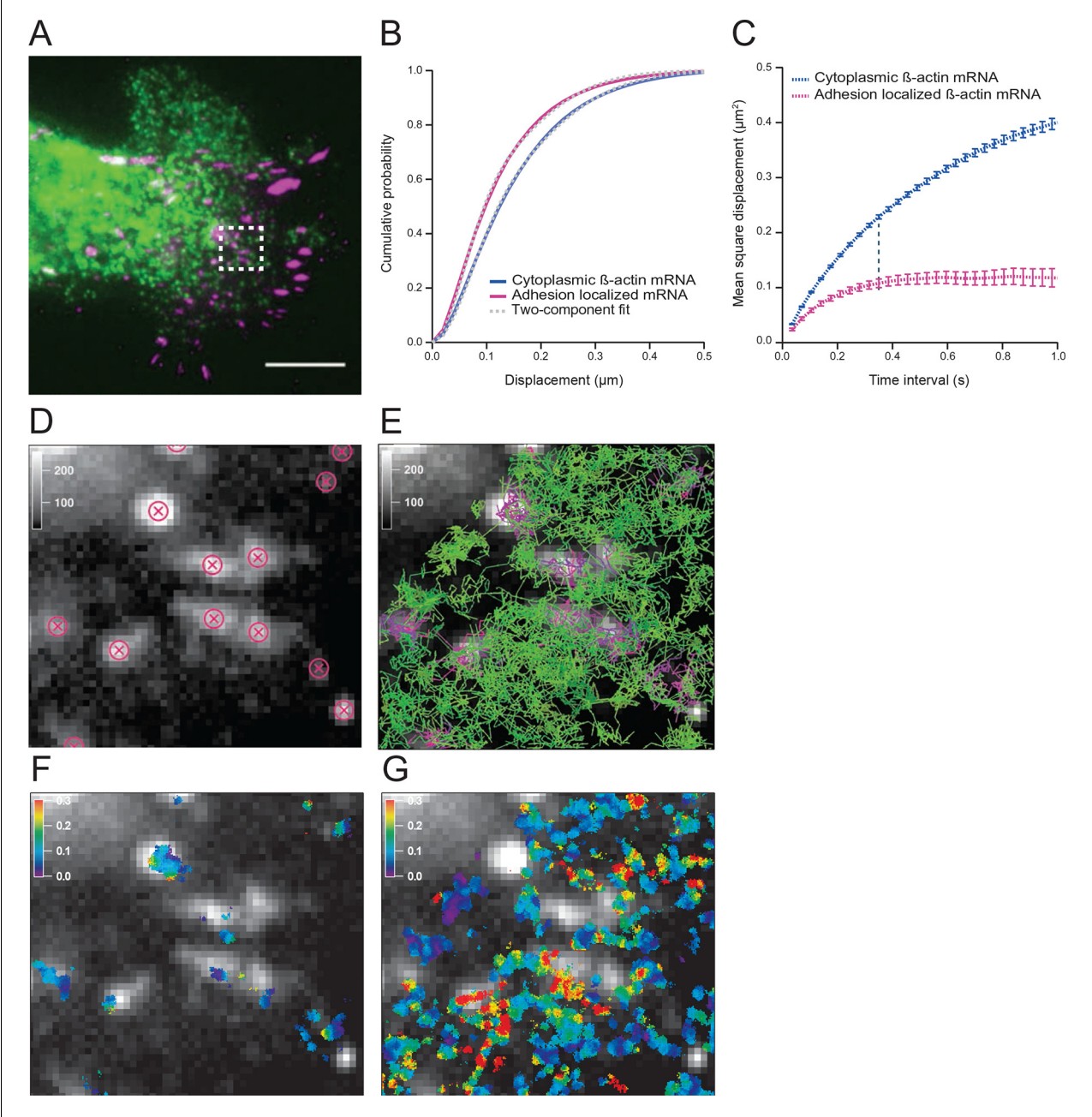

**Figure 1.** The diffusion properties of *β*-actin mRNA molecules around focal adhesions are slower and more corralled. (**A**) Overlay of a single frame from *Video 1* of β-actin mRNA labeled with MS2-GFP and focal adhesions labeled with paxillin-mCherry. (**B**) The cumulative distribution function of all single molecule displacements for β-actin mRNA localized to focal adhesions is shifted left when compared to all other mRNA, which indicates a shift towards slower diffusion. (**C**) The mean square displacement (MSD) curves depict increased corralling of mRNAs localized at focal adhesions and reveal an exploration area of 0.1 μm². (**D**) Adhesion coordinates were localized relative to the tracked mRNAs displayed in panel (**E**) in order to define a population of mRNA tracks in the vicinity of adhesions. (**F**) The mRNA diffusion heat map indicates a shift towards slower diffusion at adhesions when compared to the total cytoplasmic population of mRNA trajectories (**G**).

The following figure supplements are available for figure 1:

**Figure supplement 1.** Tracking of individual *β*-actin mRNA molecules at adhesion compartments.

**Figure supplement 2.** Cumulative distribution function analysis of *β*-actin mRNA localized to focal adhesions.

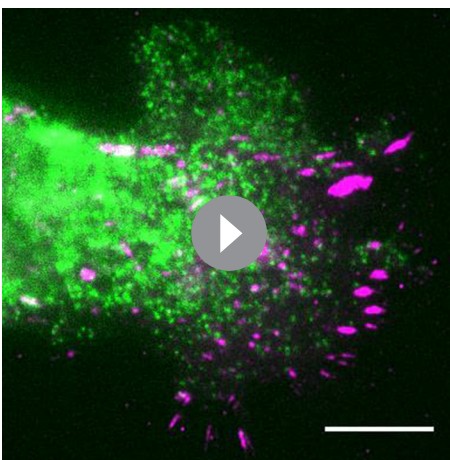

**Video 1.** Mouse embryonic fibroblasts (MEFs) from the MBS mouse with 24 MS2 stem-loop binding sites in the β-actin 3'UTR are labeled with tdMCP-GFP. Focal adhesions are labeled with paxillin-mCherry. Live cell mRNA imaging was performed with TIRF excitation for 500 streaming frames at a frame exposure time of 35 ms. The movie is played at 30 frames per second. Scale bar = 10 μm.

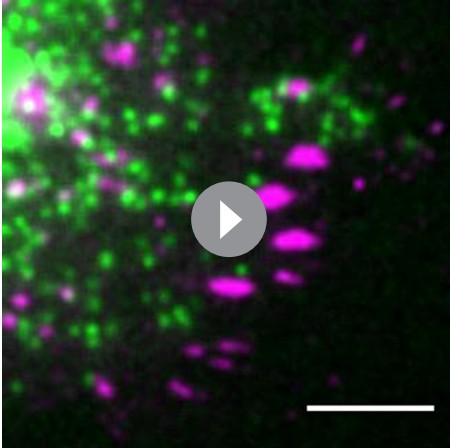

**Video 2.** The MBS MEF in *Video 1* cropped and enhanced to highlight several β-actin mRNA molecules that persist at adhesions. The scale bar is 5 μm and the movie is played at 30 fps.

diffusion was observed just as with β-actin mRNA (*Figure 2F*), and the population of faster ribosomes increased to 73% (*Figure 2—figure supplement 2B*, *Video 6*).

To demonstrate that a significant subset of ribosomes was engaged with β-actin mRNA, transcripts were tethered to adhesion plaques in live cells (depicted in *Figure 3E–G*, *Figure 3—figure supplement 1A* and *Video 7*). The adhesion protein vinculin was tagged with the MS2 capsid protein dimer (vinculin-MCP) (*Katz et al., 2012*). MBS mRNA binds to vinculin-MCP, and allowed us to observe ribosome movement at sites where β-actin mRNA is immobilized. When β-actin mRNAs are tethered to focal adhesions, ribosomes diffused significantly slower (*Figure 3—figure supplement 1B*). Tracking of ribosomes in cells with β-actin mRNA tethered to adhesions shifted the movement of ribosomes towards corralled diffusion (63% vs. 56% slow, see *Figure 2F,G*, *Figure 2—figure supplement 2A* vs. *2C*, *Figure 3G* and *Video 7*). Puromycin addition increased the movement of ribosomes in cells with mRNA tethered to adhesions (*Figure 3—figure supplement 1B* and *Video 8*). This is also the case for ribosomes in cells with non-tethered cytoplasmic β-actin mRNA (*Figure 2F, G*, and *Video 6* and *9*), suggesting that sptPALM of ribosomes in the TIRF field can identify ribosomes actively engaged and translating β-actin mRNA. In order to investigate this possibility further, we set out to perform dual-color tracking of β-actin mRNA and ribosomes. We reasoned that translating mRNAs should appear as mRNA particles moving while associated with a ribosome. Analyzing co-moving particles should therefore allow characterizing translation compartmentalization in migrating fibroblasts.

## Co-movement analysis reveals that mRNA movement is reduced and is more confined with increased ribosome load

Simultaneous TIRF excitation of GFP labeled β-actin mRNA and PATagRFP labeled ribosomes (fluorescently activated by a 405 nm laser) enabled dual-color single molecule tracking using two synchronized EMCCD cameras (*Figure 3A–B*). mRNA trajectories were spatially and temporally correlated with ribosome trajectories to identify a co-moving population for diffusion analysis (see Methods and *Figure 3—figure supplement 3*). Whereas all endogenous β-actin mRNA molecules in the cell are detected using our tdMCP-GFP labeled reporter system, only a few ribosomes (on average 10 molecules/frame) can be tracked at each given time using the sptPALM technique. Consistent with this experimental limit, only a subset of β-actin mRNA were identified as co-moving with ribosomes (*Figure 3C–D*; *Figure 4—figure supplement 3A*). We hypothesized those mRNAs that co-moved with ribosomes were likely loaded with multiple ribosomes and may exhibit slower

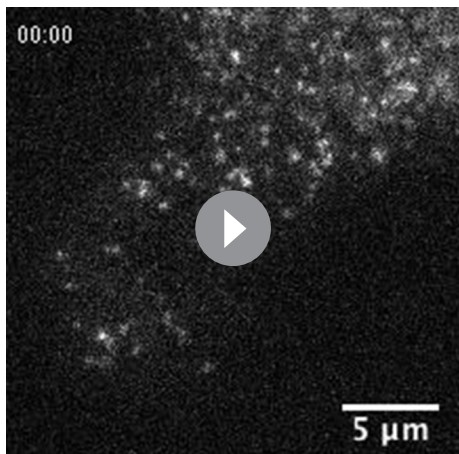

**Video 3.** β-actin mRNA is visualized 60 min after addition of puromycin. Compared to steady-state movement, mRNA without ribosomes is much faster and homogenous throughout the cytoplasm.

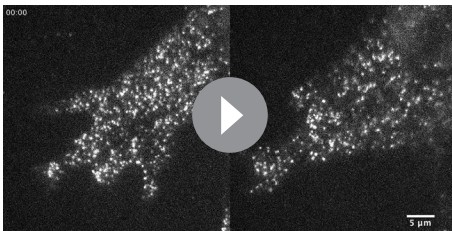

**Video 4.** β-actin mRNA is visualized 20 min after addition of hippuristanol. Compared to the control (left), the movement of mRNA without ribosomes (right, same cell 20 min later) is much faster and homogeneous throughout the cytoplasm.

movement. Consistent with this, diffusion analysis showed a shift towards slower movement in the co-moving mRNA population (*Figure 3D*; *Figure 3—figure supplement 2B*). mRNAs found to co-move with ribosomes shifted towards slower (*Figure 3—figure supplement 2A*, *Figure 4B*) and more corralled movement (*Figure 3—figure supplement 2B*), an indication that these mRNAs were in polysomes and were more likely to dwell locally. Likewise, ribosomes found to co-move with β-actin mRNA trended slower and more corralled, reminiscent of mRNA localized to focal adhesions (*Figure 2—figure supplement 2E* compared to *Figure 1C*). Together, the analysis of the mobility and co-movement of mRNA and ribosomes provide a detailed picture of the translational landscape inside live cells, capable of identifying translational hotspots (*Figure 3D,G*).

Localization of mRNAs has long been proposed to spatially regulate translation, but this has not been directly observed in live cells. In order to address this question, we set out to test previous findings showing that β-actin mRNA localized to adhesion complexes, presumably synthesizing protein to enhance focal adhesion stabilization and directed migration (*Katz et al., 2012*). In order to further investigate the localized translation near focal adhesions, expression of paxillin-mCherry in cells with mRNA and ribosomes allowed three-color colocalization analysis of co-moving trajectories near adhesions (*Video 10*). Adhesions were identified and bleached before tracking of mRNA and ribosomes (*Figure 4A*, lower panel). This allowed us to analyze all co-moving mRNAs near adhesions separately. Consistent with this prediction, a majority of ribosome-bound mRNA near the focal adhesions diffused slower than ribosome-bound mRNA away from focal adhesions (*Figure 3—figure supplement 2B*), and much slower than mRNA detected without ribosomes, where the two-component fit to the CDF revealed only 34% of slow-moving mRNAs (*Figure 4B*).

We then sought to better characterize the diffusing subpopulations using a stochastic Hidden Markov modeling (HMM)-Bayes approach that annotates motion states with single-step resolution (*Monnier et al., 2015*). HMM-Bayes fits two diffusive motion states with different diffusion coefficients and automatically annotates the individual steps of each trajectory as corresponding to either the high diffusion state (blue) or the low diffusion state (green) (see *Figure 4C* to *F* and *Figure 4—figure supplement 1* and *3*). In contrast to mean-square-displacement analysis that averages heterogeneous motion over predetermined temporal windows within each trajectory, HMM-Bayes automatically annotates stochastic motion dynamics, including the possibility of directed or active transport, without any user-supplied averaging parameters. Unlike our trajectory-averaged analysis, HMM-Bayes can provide insight into switching rates and lifetimes of the fast (D1) and slow (D2) diffusive states. It revealed that the slow diffusing state D2 is much longer lived for co-moving ribosomes (1.3 s) as compared to mRNAs that are not observed to be co-moving (0.3 s), consistent with translating ribosomes having a higher propensity to dwell at sites of translation (compare *Figure 4—figure supplement 2C* with *3C*). Interestingly, all three species have fast diffusing states D1 with similar lifetimes of ~0.5 s (*Figure 4E* and *Figure 4—figure supplement 2B* and *3B*), which may suggest

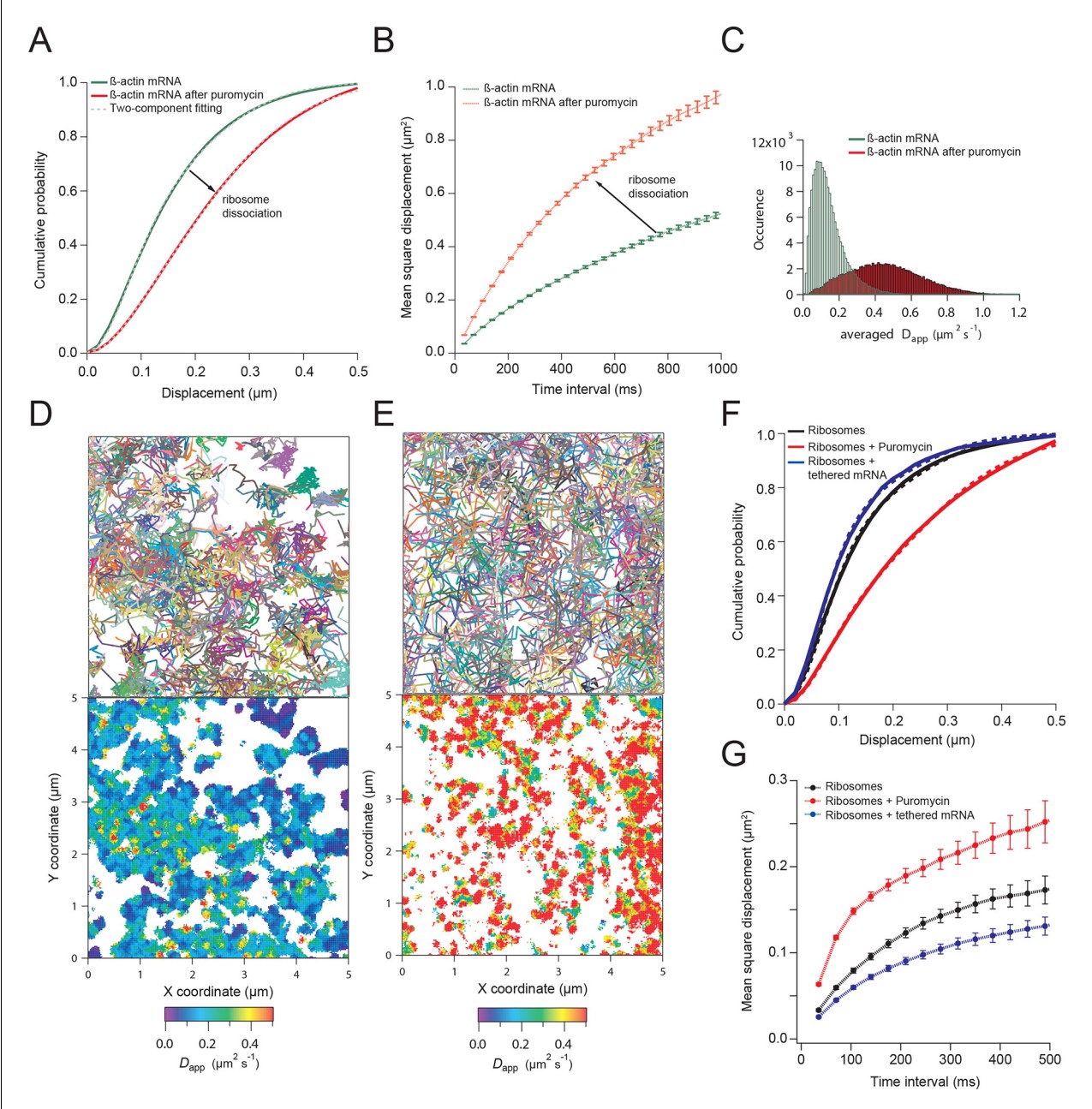

**Figure 2.** *β*-actin mRNA trajectories are faster and less-corralled after ribosome dissociation via puromycin treatment. (**A**) Treatment with puromycin dissociates ribosomes from mRNA and shifts β-actin mRNA trajectories significantly towards faster diffusion (arrow). (**B**) Mean square displacement curves also indicate a shift towards faster movement (arrow) after puromycin treatment. (**C**) The histogram of average apparent diffusion coefficients shows the transition of the population to a faster movement profile of β-actin mRNA after ribosome dissociation. (**D**) A subset of trajectories from mRNA tracking and the resulting diffusion map indicates areas of slower movement. (**E**) After puromycin treatment most trajectories shift towards faster movement as reflected in the diffusion heat map. (**F**) sptPALM of ribosomes reflects similar movement changes after puromycin treatment. Immobilizing a subset of β-actin mRNA at focal adhesions with MS2-vinculin tethering shifts the ribosome CDF curve to the left, signifying tracked ribosomes are interacting with β-actin mRNA in the adhesion compartment. (**G**) The ribosome trajectory changes after puromycin and in mRNA tethering experiments are likewise reflected in MSD curves.

The following figure supplements are available for figure 2:

**Figure supplement 1.** *β*-actin mRNA trajectories are faster and less-corralled after addition of hippuristanol.

**Figure supplement 2.** Cumulative distribution function analysis of ribosome diffusion.

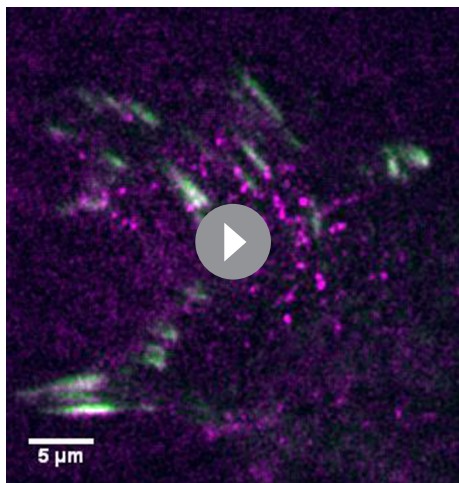

**Video 5.** Ribosomes are visualized with photo-activated localization microscopy (PALM) in fibroblasts stably expressing the L10A 60s large subunit tagged with photo-activated TagRFP (PATagRFP). 405-nm activation energy is controlled manually to limit the number of fluorescent molecules, enabling single particle tracking of ribosomes. Focal adhesions are labeled with vinculin-GFP. Ribosomes are tracked with a streaming acquisition at 35 ms per frame. The movie is played at 30 fps with a 5 µm scale bar.

that away from sites of active translation all three species explore the cytoplasmic nanoenvironment in similar ways.

As depicted in *Figure 4C* inset, ribosomes co-localize predominately with the slow diffusive mRNA state (in green) close to focal adhesions, and we hypothesize that the lower diffusion coefficient corresponds to a condition in which the mRNAs are loaded with polysomes within specific nanoenvironments of active translation.

Interestingly, lifetimes of the slow state for co-moving ribosomes and mRNAs revealed by the HMM-Bayes are transient in nature and are much shorter than the translation time: given a translation rate of 5.6 amino acids per second (*Ingolia et al., 2011*) we estimate the time to translate one β-actin protein molecule to be 67 s. The transient slow state is both more abundant and more stable for co-moving ribosomes at focal adhesions, which suggests that translating ribosomes have higher propensity to dwell within specific nanoenvironments close to focal adhesions. While the microscopic mechanism of transient translational confinement is unknown, this local ribosome confinement can be enhanced by artificially tethering β-actin mRNA to focal adhesions (*Figure 2F,G*, *Figure 3—figure supplement 1B*, *Figure 3E–G*

and *Video 7*). This indicates that local confinement of ribosomes is mediated by mRNAs that are dwelling at specific local nanoenvironments at focal adhesions. β-actin confinement close to focal adhesions has previously been shown to be functionally important for global cell locomotion (*Katz et al., 2012*), and our analysis now indicates that weak and local transient interactions mediate this confinement.

## Discussion

The MS2 labeling system was previously used to track movements of reporter RNA transfected into live cells, and demonstrated that reporter mRNAs moved through a variety of states in the cell cytoplasm, including corralled diffusion and directed motion (*Fusco et al., 2003*; *Yamagishi et al., 2009*). Using a custom microscope and novel imaging methods, we were able to acquire high temporally and spatially resolved maps of the cytoplasmic diffusion of endogenous labeled β-actin mRNA. Simultaneously imaging PATagRFP labeled ribosomes along with the labeled mRNA allowed for co-movement analysis of single mRNAs and associated ribosomes. This method of mapping mRNA translation in live cells should be widely applicable to the field of translation, which currently lacks a method for single molecule imaging of mRNAs that are on polysomes.

It has been well established that mRNA localization is a conserved mechanism utilized in multiple cell lines and species. Although it is assumed that localized mRNAs are translated, this level of regulation remains unexplored because of the lack of an appropriate method. We demonstrate an approach that can be applied to any mRNA in living cells to determine whether that mRNA is associated with polysomes at a precise location. This also allows for nanometer resolution on mRNA-protein interactions within the cell. HMM-Bayes analysis applied to the co-movement single molecule tracking data provides an additional level of information. For instance, unlike mRNA transport in the dendritic processes of neurons, active or directed transport mediated by molecular motors was not detected. We can now also identify the spatial location where individual mRNAs transition into a slower moving (translating) state. This differentiation is essential in order to investigate mechanisms

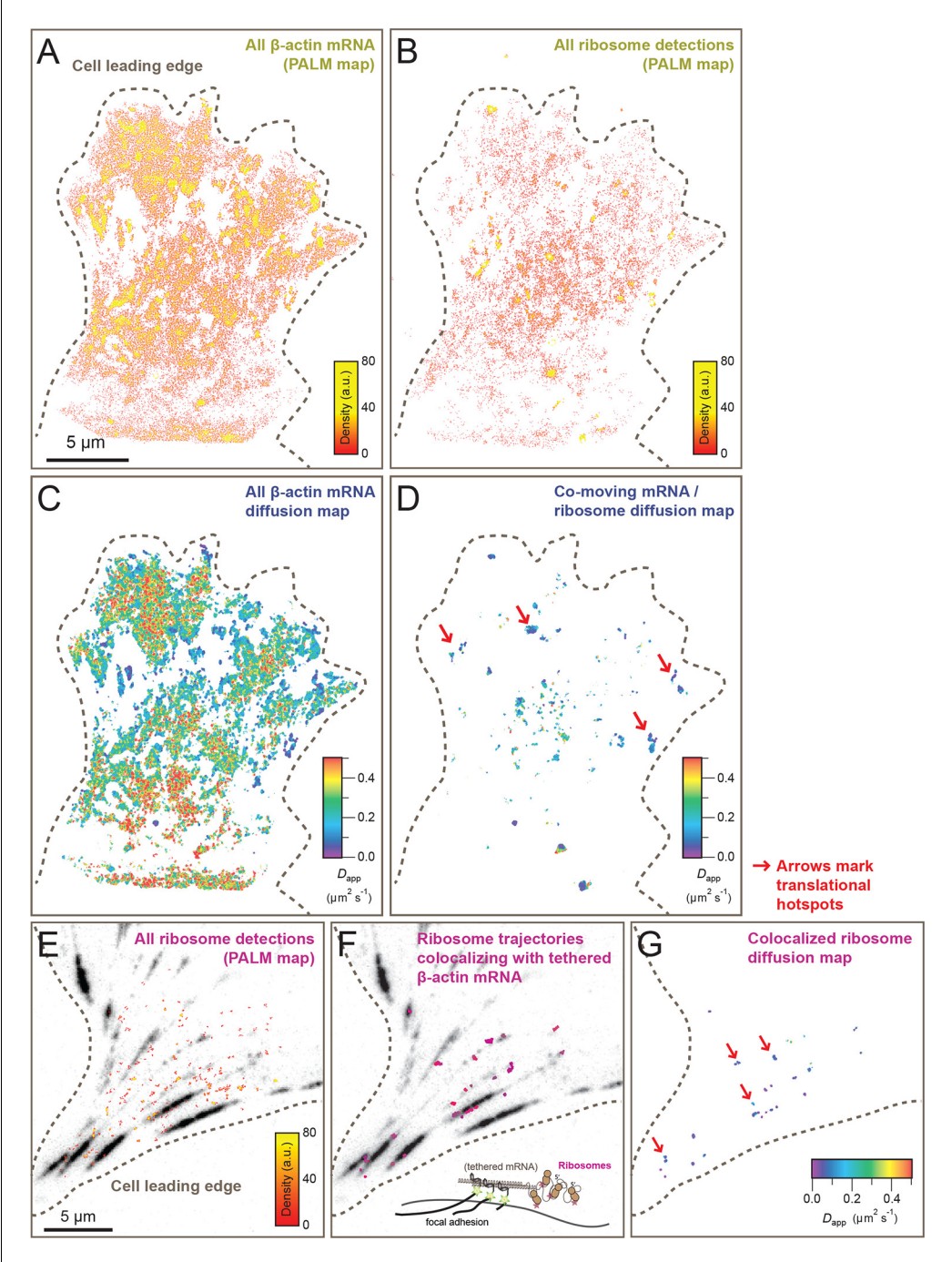

**Figure 3.** Cellular maps of areas with increased β-actin translation. (**A**) Super-resolution PALM density map composed of all 63,940 β-actin mRNA localizations from 6,235 trajectories. The PALM density plot was generated with VISP (*El Beheiry and Dahan, 2013*) from a simultaneous two-color movie with 500 consecutive frames. (**B**) Corresponding super-resolution PALM density map of all 19,420 ribosome localizations from 2,811 trajectories. (**C**) A map of the local apparent diffusion coefficients from all 6,235 β-actin mRNA trajectories. (**D**) A map of the local apparent diffusion coefficients from the 300 trajectories that were observed to be co-moving. The red arrows mark areas of enriched ribosome association with slowly diffusing β-actin mRNA. (**E**) Tracking of ribosome trajectories in cells where β-actin mRNA is tethered to focal adhesions. Super-resolution PALM density map composed of all detected localizations (1827) from 215 trajectories from 500 consecutive frames of ribosome imaging (see *Video 7*). (**F**) In shades of pink, overlay of the 33 trajectories (15.4% ) that co-localize with the focal-adhesion marker with tethered β-actin mRNA. (**G**) A map of the apparent diffusion coefficients of co-localized ribosome trajectories exhibits predominately slow diffusion characteristics when mRNA is tethered to focal adhesions. The red arrows mark the hotspots where the tethered mRNA is being actively translated.

*Figure 3 continued on next page*

*Figure 3 continued*

The following figure supplements are available for figure 3:

**Figure supplement 1.** Tracking of individual ribosome molecules with *β*-actin mRNA tethered to focal adhesions.

**Figure supplement 2.** Diffusion characteristics of *β*-actin mRNA co-moving with ribosomes and without ribosomes.

**Figure supplement 3.** mRNA/ribosome colocalization statistics in the absence and presence of puromycin.

of local regulation, for instance whether physical structures or proteins contact the molecule of interest.

HMM-Bayes analysis has revealed that individual co-moving β-actin mRNA molecules and their ribosome trajectories can interconvert between slower and faster diffusing states on the timescale of seconds. The transient slow states are enriched at microenvironments close to focal adhesions, areas known for a dense meshwork of actin fibers and actin binding proteins. The translating complex may co-translationally associate with structures like filaments, adhesion associated proteins, chaperons, enzymes or other organelles (*Condeelis and Singer, 2005*; *Kramer et al., 2009*). It is unlikely that the slow diffusing state represents a population of translating mRNAs that remain corralled for the entirety of protein synthesis since the lifetime is much shorter (a few seconds) than the time it takes to translate the actin mRNA (~ 1 min). However, the disappearance of the slow state upon ribosome disassembly indicates that the translation cycle may be involved. For instance, transitions between the slower and faster states occur on the same order as translation initiation rate (~ 0.1–0.5 s$^{-1}$) for the β-actin mRNA (*Schwanhäusser et al., 2011*). This suggests a potential connection to a single, short-lived state of the translation cycle, such as the deposition or release of initiation complex proteins. Interestingly, these functionally important yet transient interactions mediate local confinement at the cell leading edge and are reminiscent of the weak and transient nuclear interactions observed to play an important role in transcriptional initiation (*Cisse et al., 2013*). Hence, single molecule tracking is a robust method to identify spatial heterogeneity of movement both in the nucleus and cytoplasm (*Izeddin et al., 2014*; *Lu et al., 2014*; *Spille et al., 2015*).

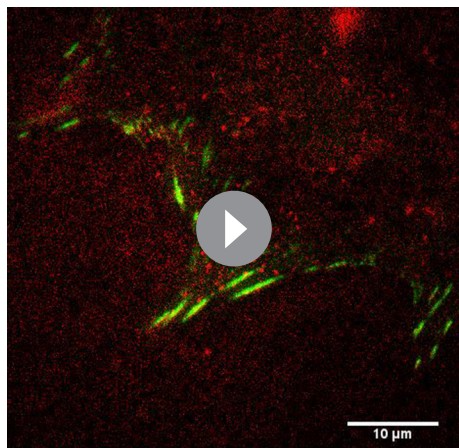

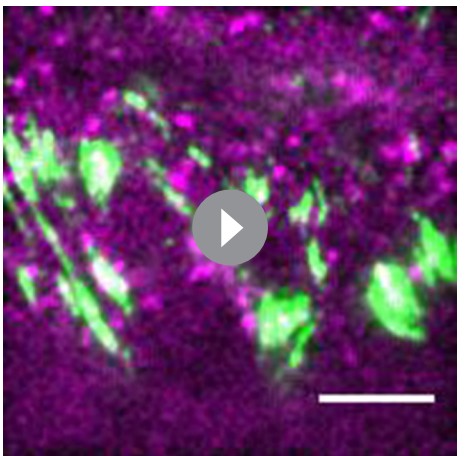

**Video 6.** Ribosomes visualized with L10A-PATagRFP after puromycin addition. Focal adhesions are labeled with vinculin-GFP. The stack was acquired at 35 ms per frame and played at 30 fps with a 5 μm scale bar.

**Video 7.** Ribosomes visualized in cells with *β*-actin mRNA tethered to focal adhesions. Non-fluorescent β-actin mRNA with MS2 stem-loops in the 3'UTR becomes tethered to vinculin that is tagged with tdMCP-GFP. Ribosomes can be seen preferentially immobilizing at larger focal adhesions that have accumulated mRNA over time. Tethering mRNA to adhesions significantly slows the average diffusion of ribosomes and therefore shows a significant population of tracked ribosomes are interacting with β-actin mRNA. The stack was acquired at 35 ms per frame and played at 30 fps with a 10 μm scale bar.

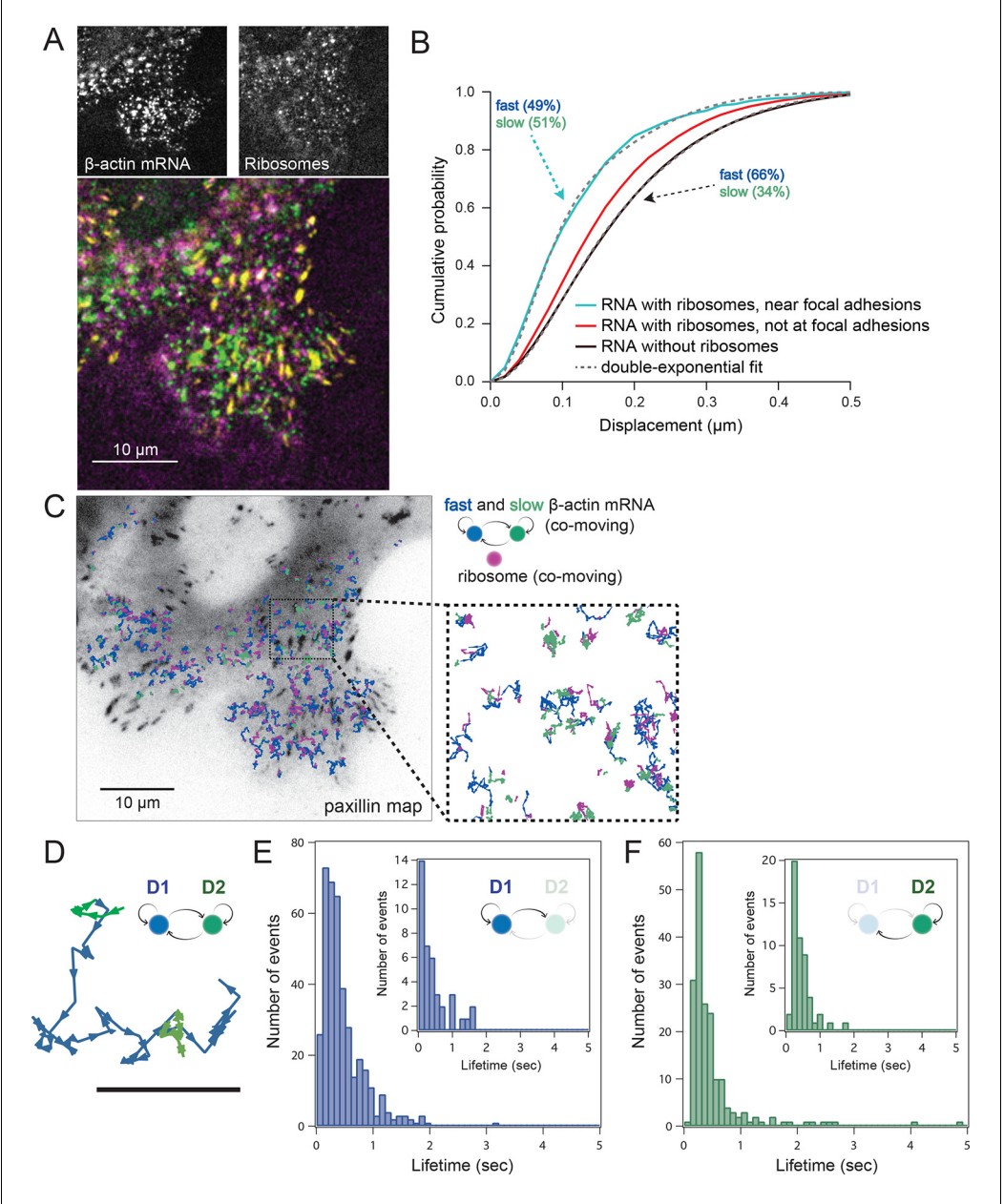

**Figure 4.** Multi-color live cell imaging and simultaneous tracking of β-actin mRNA and ribosomes at focal adhesions. (**A**) The image from a single frame from *Video 10* depicts simultaneous tracking of β-actin mRNA (green) and ribosomes (magenta) in live mouse embryonic fibroblasts with focal adhesions labeled with paxillin-mCherry (yellow). The adhesions were bleached prior to tracking mRNA and ribosomes. (**B**) Compartmentalized co-movement analysis of β-actin mRNA and ribosomes elucidates a shift in the displacement towards slower diffusion of mRNA visualized with ribosomes at adhesions. (**C**) β-actin mRNA trajectories were assessed by HMM-Bayesian analysis to identify diffusive state switching (fast = blue, slow = green) relative to ribosome trajectories (magenta) near adhesion proteins mapped in black. (**D**) One co-moving β-actin mRNA trajectory from **C**) is displayed at higher magnification. HMM-Bayesian analysis clearly visualizes that the trajectory interconverts between the D1 and D2 states multiple times. Scale bar: 1 μm. (**E**) State-duration histogram for all fast states D1 as determined from all co-moving β-actin mRNA trajectories that sampled the state D1 (from 382 state visits by co-moving mRNA trajectories). Insert: Histogram of only internal state durations of D1 (excluding states at start and end of each track). (**F**) Corresponding state-duration histogram of all the slow state (D2) durations as determined from 187 visits by co-moving β-actin mRNA trajectories to the state D2. The insert depicts the internal state durations (excluding states at start and end of each track) for D2.

The following figure supplements are available for figure 4:

**Figure supplement 1.** HMM-Bayesian analysis of co-moving mRNAs.

*Figure 4 continued*

**Figure supplement 2.** HMM-Bayesian analysis of co-moving ribosomes.

**Figure supplement 3.** HMM-Bayesian analysis of non-co-moving mRNAs.

Understanding the frequency of state transitions and their local nanoenvironment will be essential in future live cell imaging studies.

In addition, the method of mRNA tracking and mapping described here could by tagging another mRNA characterize multiple mRNA targets that may be localized and regulated in the same manner as β-actin mRNA. Since there are additional RNA labeling systems, like PP7 stem loops which work orthogonally to MS2, it is now reasonable to test the dynamics of translational regulation for many of the reported target mRNAs (*Hocine et al., 2013*). New results from microarray data, deep sequencing, and other biochemical approaches will require live cell tracking approaches to experimentally validate where and when these mRNAs localize (*Poon et al., 2006*; *Sutton and Schuman, 2006*; *Patel et al., 2012*). Development of sptPALM probes with greater photostability will enable longer-term imaging concomitant with MS2-based labeling, and future advances in imaging systems that can simultaneously track single molecules in three channels will allow any RNA of interest to be characterized. The PATagRFP-L10A ribosome-tracking construct can be used to map co-movement and diffusion characteristics of any uncharacterized mRNA. In particular, it would be advantageous to simultaneously track and map mRNA and ribosomes in neurons where local translation at the synapse is thought to contribute to information processing. Importantly, once the diffusion characteristics of mRNA in polysomes have been defined, it will not be necessary to use labeled ribosomes to verify that translation is occurring. Conversely, mapping ribosome diffusion in areas of the cell would reveal regions of higher translation. Therefore assessing diffusion coefficients using single particle tracking will provide new detailed information on translation.

We have previously observed that β-actin mRNA localization at the leading edge of cells is highly correlated with motility (*Park et al., 2012*). This localization appears to surround the focal adhesion complexes (FACs) nearest to the lamellipod (*Katz et al., 2012*). By tethering the β-actin mRNA to all

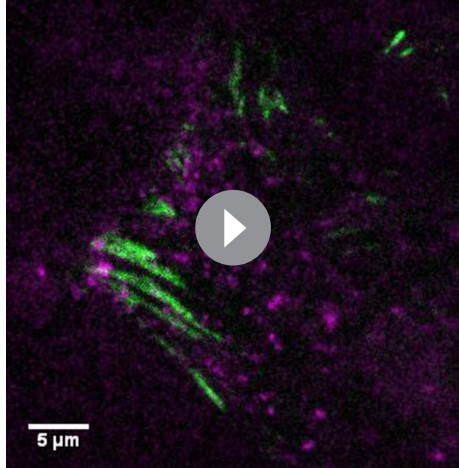

**Video 8.** Ribosomes in cells with *β*-actin mRNA tethered to adhesions after the addition of puromycin. Compared to the ribosomes in *Video 7*, it is easy to see that puromycin dissociates ribosomes from mRNA tethered at focal adhesions. The stack was acquired at 35 ms per frame and played at 30 fps with a 5 μm scale bar.

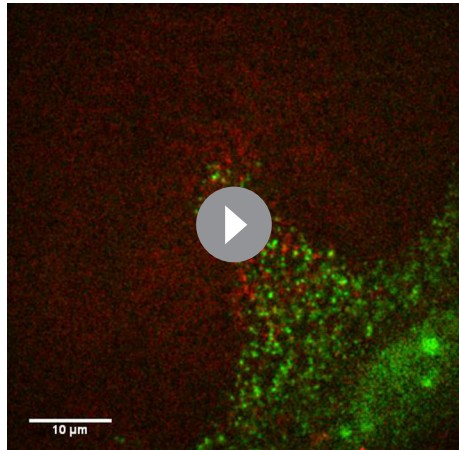

**Video 9.** *β*-actin mRNA and ribosomes are simultaneously visualized and tracked after puromycin addition. The stack was acquired at 35 ms per frame and played at 30 fps with a 10 μm scale bar.

FACs indiscriminately to vinculin we inhibit cellular motility since mRNA localization cannot generate any asymmetry among the focal adhesions (*Katz et al., 2012*). We have shown that β-actin is preferentially translated in the leading edge (*Wu et al., 2015*). Here, we have investigated the translation of individual β-actin mRNAs by tracking their co-movement with ribosomes at high spatiotemporal resolution in live cells. The results corroborate and extend the hypothesis that actin translation in general is preferred around the focal adhesion complex. This localized translation could generate a more concentrated supply of not only actin, but also the many proteins associated with focal complex formation (*Kanchanawong et al., 2010*), the proteins that nucleate actin such as Arp2/3 (*Mingle et al., 2005*), and possibly FAC proteins as well. Hence the assembly of the complex starting with integrin association may be aided co-translationally, promoting 'waves' of actin polymerization (*Case and Waterman, 2011*; *Case and Waterman, 2015*). Therefore, we suggest that models of cell motility (*Danuser et al., 2013*) should consider the role of localized RNPs in facilitating cell motility by assembly of multipolypeptide complexes at adhesion sites.

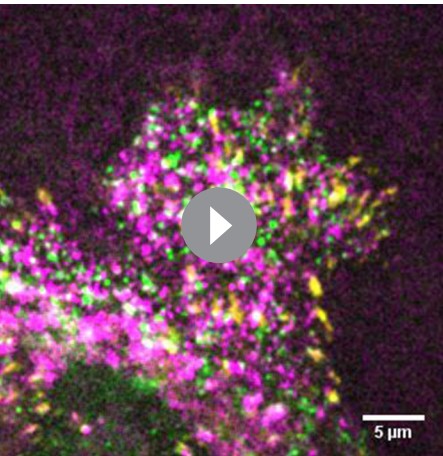

**Video 10.** β-actin mRNA and ribosomes are simultaneously visualized and tracked after the cell's focal adhesions were identified with paxillin-mCherry and bleached. The stack was acquired at 35 ms per frame and played at 30 fps with a 5 µm scale bar.

## Materials and methods

### Cell lines and reagents

Mouse embryonic fibroblasts (MEFs) derived from the MS2 β-actin knock-in mouse (*Lionnet et al., 2011*) were used for tracking endogenous β-actin mRNA as described previously. Briefly, cells were isolated from stage E14.5 embryos and immortalized after transformation by transfecting primary cells with SV40 large T-antigen. Focal adhesion label paxillin-mCherry was transiently expressed from pcDNA 3.1 plasmid after nucleofection (Amaxa Biosystems, Gaithersburg, MD) and imaged 24–48 hr later. The MS2 stem loops in the 3'UTR of endogenous β-actin mRNA were labeled with 2xMCP-GFP stably integrated into the cells by lentiviral infection. PATagRFP-L10A was also integrated through lentiviral infection. Cells were grown in DMEM high glucose with 10% FBS, penicillin and streptomycin. MEFs were imaged on MatTek (Ashland, MA) dishes or Bioptechs (Butler, PA) dishes and imaged in Leibovitz's L15 medium (without Phenol Red, Gibco, Gaithersburg, MD) with 10% FBS. Glass coverslips were coated with 10 µg/ml fibronectin (Sigma-Aldrich, St. Louis, MO) 30 min before plating cells for imaging experiments. Puromycin (Sigma-Aldrich) was added at 100 µg/ml to disrupt translation and hippuristanol (gift from Dr. Jerry Pelletier, McGill University) was added at 1 µM to disrupt translation initiation.

### Image acquisition

MEFs were imaged on a previously described custom-built microscope (*Grünwald and Singer, 2010*; *Katz et al., 2012*). Simultaneous dual-color imaging experiments of mRNA and ribosomes were performed with an Olympus (Waltham, MA) 1.45 NA 150X objective with a resulting pixel size of 107 nm. Cobolt (San Jose, CA) Jive 50TM 561 nm and Cobolt Calypso 100 491-nm lasers were coupled into a single optical fiber, collimated, and delivered through the back port of the Olympus IX-71 stand. A tunable lens was inserted into the light path to produce objective base total internal reflection fluorescent excitation. A 405 nm laser (Coherent, Santa Clara, CA) was free-space-coupled into the back port of the microscope stand for full field photo-activation. Activation was manually controlled with a UNIBLITZ shutter (Vincent Associates, Rochester, NY) and purposely kept low (~1.5% of total possible output) so only a few ribosomes were fluorescently activated for single

particle tracking. Imaging experiments were run through MetaMorph Software (Molecular Devices, Sunnyvale, CA).

## Image analysis

16-bit Tiff stacks for each channel were exported from MetaMorph to DiaTrack software (v. 3.03, Semasopht, Switzerland), which identifies and fits the intensity spots of fluorescent particles with 2D Gaussian functions matched to the experimentally determined PSF. Images were background-sub- tracted and processed with a Gaussian filter of a half width at half max value of 1.3 pixels, which was user-determined to be the best value for particle identification. Thresholds for particle selection were determined for each data series. Care was taken to observe at least 50-100 frames of process- ing to ensure most particles were identified and there was no incidence of false detections due to background. The data were then tracked with a maximum particle displacement of 5 pixels (~533 nm) between points in a track. Tracks with a lifetime of 3 frames or more were exported for co- movement analysis and 2D diffusion mapping.

Diffusion mapping was performed with code written in Igor 6.2 (WaveMetrics, Portland, OR). Briefly, the program extracted the trajectories from DiaTrack analysis and calculated local apparent diffusion coefficients evaluated in 20 nm x 20 nm grids from the mean square displacements over a timescale of 35 ms. Whenever five or more separate displacements originating within 80 nm of a given grid node were found, a local apparent diffusion coefficient was calculated and plotted. A heat map from all trajectories observed during the entire acquisition period was created to repre- sent the average local apparent diffusion coefficient. All trajectories were also analyzed by calculat- ing and fitting cumulative distribution functions (CDFs) of displacements to determine the average diffusion coefficient. The average mean square displacement (MSD) curve for all trajectories was determined over different time intervals. Error bars represent the experimental standard errors of the means.

Co-movement analysis was performed in MatLab (MathWorks, Natick, MA). Both RNA and ribo- some stacks were tracked using the commercial tracking software DiaTrack (v. 3.03) for single parti- cle tracking analysis. Each trajectory coordinate was time encoded so tracks that moved within 320 nm (3 pixels) of each other for at least 105 ms (3 frames) were considered co-moving. A list of co- moving trajectories for each dataset were saved and input into the custom analysis routines written in Igor Pro for diffusion analysis and mapping.

In our experiments all β-actin mRNAs are detected. However, translation events are only observed when a β-actin mRNA is translated by a ribosome containing L10A-PATagRFP, and when the fluorescence of this PATagRFP molecule is activated by our violet laser. Two potential concerns arise: (1) Only a small unrepresentative fraction of ribosomes might contain the PATagRFP-L10A. This is not the case: in a previous publication from our group (*Wu et al., 2015*), we showed that labeled L10A is successfully incorporated into ~40% of the total ribosome population. (2) Only low numbers of PATagRFP-L10A molecules can be fluorescently activated at any given time. Since we must limit our photoactivation laser powers to a regime where we avoid overlap between individual ribosome molecules, we cannot observe all ribosomes simultaneously and this prevents us from observing the absolute number of ribosomes per mRNA, or any other metric that would depend on the degree of labeling of the various species. We first ensured through an ad-hoc co-moving detec- tion algorithm that all co-movement events were stringently selected (the algorithm was introduced by our group in an earlier publication: *Halstead et al., 2015*). This is important because false posi- tives could affect our mobility measurements. Biologically specific co-moving events are apparent when looking at the pooled distances between mRNAs and ribosomes (see *Figure 3—figure sup- plement 3*). High stringency runs the risk of detecting too few trajectories to build robust statistics. We ensured this is the case by acquiring vast numbers of trajectories: in a typical experiment, we would detect ~9000 individual mRNA trajectories, out of which ~5% passed the co-moving test. This represents ~450 co-moving events, from a single movie. The aggregate data for co-movement analy- sis come from many movies in different cells, and therefore represent thousands of co-moving events.

The trajectories were also analyzed with HMM-Bayes, which infers diffusive and directed motion states from observed particle displacements and annotates when and where each motion state occurs in space and time along each trajectory with single-step resolution (*Monnier et al., 2015*). HMM-Bayes analysis was performed on pooled sets of trajectories; thus the motion parameters

(diffusion coefficients) were inferred considering all trajectories together, but state sequence annotations were inferred separately for each trajectory. Motion state lifetimes were extracted from the inferred sequence of motion states along each trajectory, either including or excluding the durations of the states at the beginning and end of each trajectory as noted.

## Additional information

### Competing interests

RHS: Reviewing editor, *eLife.* The other authors declare that no competing interests exist.

### Funding

| Funder | Grant reference number | Author |
|---|---|---|
| National Institutes of Health | NS083085 | Zachary B Katz<br>Brian P English<br>Young J Yoon<br>Robert H Singer |
| Howard Hughes Medical Institute | Janelia Research Campus | Robert H Singer<br>Brian P English |
| NSF Office of the Director | PoLS PHY 1305537 | Nilah Monnier<br>Mark Bathe |
| National Institutes of Health | R01 GM076293 | Ben Ovryn |

The funders had no role in study design, data collection and interpretation, or the decision to submit the work for publication.

### Author contributions

ZBK, BPE, Conception and design, Acquisition of data, Analysis and interpretation of data, Drafting or revising the article; TL, BO, RHS, Conception and design, Analysis and interpretation of data, Drafting or revising the article; YJY, Conception and design; NM, Analysis and interpretation of data; MB, Analysis and interpretation of data, Drafting or revising the article

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
