## [Decision Letter]

Thank you for submitting your work entitled "Mapping translation in live cells by tracking single molecules of mRNA and ribosomes" for consideration by *eLife*. Your article has been reviewed by two peer reviewers, and the evaluation has been overseen by a Reviewing Editor and James Manley as the Senior Editor.

The reviewers have discussed the reviews with one another and the Reviewing editor has drafted this decision to help you prepare a revised submission.

Critical points:

The two reviewers agree that your paper present a significant technological advance and is in principle worthy of publication in *eLife*. However, they also raised criticism of the use of puromycin, because it has non-specific effects and does not strip ribosomes rapidly and effectively from mRNA. We therefore suggest that you use a second drug that strips ribosomes from messages. Probably an initiation inhibitor like hippuristanol would be best. We would also like to see confirmation that the drug is in fact collapsing polysomes. We also believe that a much more rigorous discussion of your ribosome/mRNA colocalization statistics is required. The worry is that you are seeing an artifactual subpopulation. This together with a more robust discussion/analysis should not be too burdensome and would greatly strengthen the manuscript.

Below are listed the points made by the reviewers, which include additional comments that you should consider.

*Reviewer #1:*

There is increasing evidence pointing to a central role of localized protein translation in controlling the site and timing of protein production. However, strategies for monitoring localized translation in vivo are just now being developed. Localized translation of actin at the leading edge of motile cells has served as a powerful model system for studying these phenomena. Elegant work from the Singer lab has used MS2 coat protein-GFP fusions together with arrays of MS2 binding sites inserted into the 3'UTR of the actin message to visualize individual actin messages in a live cell. These studies have revealed different dynamics of the actin message depending on its location as well as the presence of two distinct sub populations one with faster and a second with a slower diffusion constant. In the present study, the authors develop an approach for monitoring translation of the MS2-GFP actin messages. Specifically they argue that messages being translated show significantly different kinetics of translation than non-translating messages and thus the diffusion properties can be used to report on the translation state of a labeled message. While it is not clear that there are major new functional insights directly emerging from the present dataset (beyond the important conclusion that translation impacts mRNA dynamics), the approach does represent a technical tour de force and gets at information that would otherwise be very difficult to obtain. The authors' connection between translation and the diffusion properties of a message is based on a number of highly suggestive correlations. For example, they show that messages with correlated movement with large ribosomal subunits (and thus are presumably being translated) show slower movement. Additionally, an inhibitor of translation (puromycin) leads to enhanced actin message movement and localizing the actin message to a region of high translation also slow its dynamics.

My major concern is that these arguments are correlative and qualitative in nature and each has significant caveats. This is particularly concerning as the rate of conversion between the slow and fast state is very rapid and doesn't correlate with any obvious feature of translation.

1) Puromycin is a sledge hammer that broadly impacts the cell. It would be much more compelling if the authors could specifically inhibit translation of the actin message. This should be relatively straightforward with CRIPSR engineering approaches (for example, an in frame stop codon just downstream of the start could be introduced).

2) There should be some effort to make a quantitative measures relating the fraction of actin messages that are well translated (e.g., by polysome profiling) with the kinetic properties of the different populations of actin messages. Along this lines, there is no guarantee that puromycin will cause full release of messages from ribosomes in vivo. Basically, puromycin release is competing with ongoing initiation which is quite rapid (up to 0.1/sec) so one often see high ribosome density at the 5' end of open reading frames which trails of later in the messages as ribosome have more time to be acted on by puromycin.

3) It would not be surprising if localizing any message to the adhesions sites (independent of any changes in translation) could impact the diffusion kinetics of that message. A good control would be to eliminate the translation control of actin (perhaps by knocking down the zip code protein) or localize another message to local adhesion and monitor how this impacts dynamics.

*Reviewer #2:*

The manuscript of Katz et al. uses fluorescence microscopy methods to track mRNAs and ribosomes simultaneously in vivo. To my knowledge, this is the first in vivo examination of the coupled behavior of ribosomes and mRNAs, and as such, deserves publication in *eLife*. The authors use their developed methods to examine mRNA movement in mouse cells; they have previously engineered an actin gene with multiple MS2 coat protein binding sites in the 3'UTR, allowing binding of an expressed fluorescent MS2 coat in high copy for sensitivity. Using this mRNA, they can track temporally the spatial localization of the mRNA. They have done this previously, and noted different populations of mRNAs with different diffusion times. Here they use a second color in their microscopy, and by labeling ribosomal large subunits, can correlate mRNAs with ribosomes. Their results show quite clearly that ribosome occupancy is correlated with slower diffusion and localization to focal points where actin synthesis is high. The data are rigorously analyzed (and quite nicely using multiple models including a sophisticated HMM-Baysian approach. In short, I really liked this manuscript.

My only concern is that the overall co-localization of ribosomes and mRNA appear statistically small-part of this is technical, due to the excitation of the ribosomal fluorophores, but the authors should more clearly expound on ribosomal labeling efficiency and the relevance of their colocalization numbers. Since I view this manuscript as foundational to many future studies, this change is essential.

---

## [Author Response]

*The two reviewers agree that your paper present a significant technological advance and is in principle worthy of publication in eLife. However, they also raised criticism of the use of puromycin, because it has non-specific effects and does not strip ribosomes rapidly and effectively from mRNA. We therefore suggest that you use a second drug that strips ribosomes from messages. Probably an initiation inhibitor like hippuristanol would be best. We would also like to see confirmation that the drug is in fact collapsing polysomes.*

We appreciate the thoughtful comments from the reviewers. To address the main concern towards puromycin we performed experiments with hippuristanol as suggested. An additional supplementary figure (Figure 2—figure supplement 1) and video (Video 4) with references in the main text highlight these results, which supported the puromycin experiments. We have supplied additional references in the Results section from previously published works (both our own and other groups) that confirmed the drug disrupts mRNA-ribosome interactions and collapses polysomes.

*We also believe that a much more rigorous discussion of your ribosome/mRNA colocalization statistics is required. The worry is that you are seeing an artifactual subpopulation. This together with a more robust discussion/analysis should not be too burdensome and would greatly strengthen the manuscript.*

We have added a paragraph on colocalization statistics to the Image Analysis Materials and methods section and added Figure 3:

“In our experiments all β-actin mRNAs are detected. However, translation events are only observed when a β-actin mRNA is translated by a ribosome containing L10A-PATagRFP, and when the fluorescence of this PATagRFP molecule is activated by our violet laser. […] This represents ~450 co-moving events, from a single movie. The aggregate data for co-movement analysis come from many movies in different cells, and therefore represent thousands of co-moving events.”

This is sufficient to obtain statistically significant results, as observed with the HMM-Bayes approach, which agnostically verified our two subpopulations.

Furthermore, we note that we made no claims regarding the absolute number of ribosomes per mRNA or any other metric that would depend on the degree of labeling of the various species. We limited our analysis to mobility properties that are indifferent to the fraction of molecules detected.

*Below are listed the points made by the reviewers, which include additional comments that you should consider.*

Reviewer #1:

[…] My major concern is that these arguments are correlative and qualitative in nature and each has significant caveats. This is particularly concerning as the rate of conversion between the slow and fast state is very rapid and doesn't correlate with any obvious feature of translation.

*1) Puromycin is a sledge hammer that broadly impacts the cell. It would be much more compelling if the authors could specifically inhibit translation of the actin message. This should be relatively straightforward with CRIPSR engineering approaches (for example, an in frame stop codon just downstream of the start could be introduced).*

We understand the concern towards using puromycin and, per the Reviewing editor and reviewers suggestion, we performed experiments with hippuristanol (see above).

*2) There should be some effort to make a quantitative measures relating the fraction of actin messages that are well translated (e.g., by polysome profiling) with the kinetic properties of the different populations of actin messages. Along this lines, there is no guarantee that puromycin will cause full release of messages from ribosomes in vivo. Basically, puromycin release is competing with ongoing initiation which is quite rapid (up to 0.1/sec) so one often see high ribosome density at the 5' end of open reading frames which trails of later in the messages as ribosome have more time to be acted on by puromycin.*

We added citations in the text to a recent publication that uses FCS to quantify mRNA-protein interactions, including ribosome binding at two different cellular locations. These cell lines were prepared in the same way as our study, only with mCherry-L10A labeled ribosomes (see Wu, B. et al. "Quantifying Protein-mRNA Interactions in Single Live Cells." Cell 162(1): 211-220.). Since FCS localizes its detection volume in only a single cellular location it is not capable of detecting localized translation hot-spots as with the tracking method we describe here. Blocking translation initiation with hippuristanol ensured messages were released from ribosomes and similar results were observed compared to puromycin experiments (see Figure 2—figure supplement 1).

*3) It would not be surprising if localizing any message to the adhesions sites (independent of any changes in translation) could impact the diffusion kinetics of that message. A good control would be to eliminate the translation control of actin (perhaps by knocking down the zip code protein) or localize another message to local adhesion and monitor how this impacts dynamics.*

We used the ZBP1 Knock Out cells the reviewer suggests in a previous study. We indeed observed a decrease in dwelling/corralled mRNA (i.e. highly translating mRNA) near adhesion complexes in ZB1 KO lines (see Katz, Z. B., et al. 2012). We therefore did not repeat this experiment here, but mentioned this important reference.

Reviewer #2:

*The manuscript of Katz et al. uses fluorescence microscopy methods to track mRNAs and ribosomes simultaneously in vivo. To my knowledge, this is the first in vivo examination of the coupled behavior of ribosomes and mRNAs, and as such, deserves publication in eLife. The authors use their developed methods to examine mRNA movement in mouse cells; they have previously engineered an actin gene with multiple MS2 coat protein binding sites in the 3'UTR, allowing binding of an expressed fluorescent MS2 coat in high copy for sensitivity. Using this mRNA, they can track temporally the spatial localization of the mRNA. They have done this previously, and noted different populations of mRNAs with different diffusion times. Here they use a second color in their microscopy, and by labeling ribosomal large subunits, can correlate mRNAs with ribosomes. Their results show quite clearly that ribosome occupancy is correlated with slower diffusion and localization to focal points where actin synthesis is high. The data are rigorously analyzed (and quite nicely using multiple models including a sophisticated HMM-Baysian approach. In short, I really liked this manuscript.*

*My only concern is that the overall co-localization of ribosomes and mRNA appear statistically small-part of this is technical, due to the excitation of the ribosomal fluorophores, but the authors should more clearly expound on ribosomal labeling efficiency and the relevance of their colocalization numbers. Since I view this manuscript as foundational to many future studies, this change is essential.*

We have revised the third paragraph of the Discussion to posit multiple possibilities for the transitions observed between a slow and fast state of movement for β-actin mRNA. We have also altered our estimation of translation initiation. We note that the data to estimate this value was based on a population study (Schwanhausser et al., Nature 2011) and therefore the quoted number might be different from our single-molecule data: their values were calculated based on total populations of proteins and mRNAs, which would include non-translating species such as nuclear mRNAs.

We also added Figure 3—figure supplement 3 that addresses this important point referenced in the Methods and Results sections.